# A Review of Artificial Intelligence in the Rupture Risk Assessment of Intracranial Aneurysms: Applications and Challenges

**DOI:** 10.3390/brainsci13071056

**Published:** 2023-07-11

**Authors:** Xiaopeng Li, Lang Zeng, Xuanzhen Lu, Kun Chen, Maling Yu, Baofeng Wang, Min Zhao

**Affiliations:** 1Department of Neurosurgery, Tongji Hospital, Tongji Medical College, Huazhong University of Science and Technology, Wuhan 430030, China; lixiaopeng@tjh.tjmu.edu.cn (X.L.); gdzenglang@163.com (L.Z.); kchen1722@163.com (K.C.); 2Department of Neurology, The Third Hospital of Wuhan, Wuhan 430074, China; gdzengl@yahoo.com (X.L.); coney0522@163.com (M.Y.)

**Keywords:** intracranial aneurysms, machine learning, deep learning, artificial intelligence, risk assessment

## Abstract

Intracranial aneurysms (IAs) are highly prevalent in the population, and their rupture poses a significant risk of death or disability. However, the treatment of aneurysms, whether through interventional embolization or craniotomy clipping surgery, is not always safe and carries a certain proportion of morbidity and mortality. Therefore, early detection and prompt intervention of IAs with a high risk of rupture is of notable clinical significance. Moreover, accurately predicting aneurysms that are likely to remain stable can help avoid the risks and costs of over-intervention, which also has considerable social significance. Recent advances in artificial intelligence (AI) technology offer promising strategies to assist clinical trials. This review will discuss the state-of-the-art AI applications for assessing the rupture risk of IAs, with a focus on achievements, challenges, and potential opportunities.

## 1. Introduction

Intracranial aneurysms (IAs) remain a significant public health issue, with a prevalence of up to 3.2% [1]. Incidentally detected aneurysms are becoming more common due to the availability and increasing quality of non-invasive imaging techniques [1]. Unruptured intracranial aneurysms (UIAs) can either remain stable for extended periods or grow rapidly and rupture within a short time [2,3].

IAs constitute the predominant cause of spontaneous subarachnoid hemorrhage (sSAH) and are responsible for approximately 85% of all cases of SAH [4]. IAs have a high mortality rate of 23–51%, and they carry a significant risk of causing permanent disability, which occurs in 10–20% of cases [4,5,6]. However, it should be noted that the treatment of aneurysms is not without risks. Both interventional embolization and craniotomy clipping surgery carry the inherent risk of invasion, with a post-operative morbidity incidence ranging from 4.3–4.6%. Moreover, treatment may lead to new neurological deficits, with an incidence rate of 10–24.6% [7,8]. Therefore, early detection of IAs with high risk of rupture and prompt intervention to avoid their catastrophic consequences have notable clinical significance to physicians. Meanwhile, the risks and expenses of over-intervention can also be avoided with accurate prediction of aneurysms that may remain stable for a long time, which also has important socioeconomic implications.

The management of IAs presents a significant challenge for physicians and patients alike, whether the aneurysm is discovered incidentally or during screening. The decision to intervene requires a careful consideration of the periprocedural risks associated with endovascular or surgical procedures, as well as the risk of subarachnoid hemorrhage. Thus, addressing this issue using conventional methodologies is often challenging. Taking small unruptured aneurysms as an example, management strategies for small UIAs (<7 mm) demonstrate that there are noticeable variations. Malhotra et al. [9] presented a survey of 227 active neurointerventionalists and neuroradiologists, wherein the frequency and manner of follow-up imaging for unruptured cerebral aneurysms were found to vary considerably in clinical practice. Of the 227 respondents, 174 indicated that regular and periodic imaging surveillance was appropriate for the conservative management of unruptured cerebral aneurysms. Furthermore, 84% of the respondents recommended a surveillance frequency of at least once a year. Regarding follow-up imaging modality, 59% of the respondents preferred noncontrast MR angiography. Additionally, a similar number of respondents advocated for indefinite, lifelong follow-up for minor UIAs. Both the criteria employed to determine growth on surveillance imaging and the size measures used to evaluate aneurysms exhibited significant variability.

It is necessary to use techniques and models that can enhance overall suggestions due to the variety and variability in practice. Recently, researchers and medical professionals have begun to pay more attention to Artificial Intelligence (AI) in relation to this issue. AI pertains to the development of computer algorithms that replicate human cognitive abilities such as learning, reasoning, and self-correction. The advancement of machine-based analysis of intricate data for routine clinical use has demonstrated remarkable progress [10]. Present-day image identification and risk assessment of IAs are two areas where AI technology has considerable potential for advancement. Additionally, it is anticipated to meet the clinician’s demands for greater efficacy and precision when evaluating the rupture probability of cerebral aneurysms [11,12].

This review focuses on how AI is currently used to estimate the likelihood of a cerebral aneurysm rupturing, as well as exploring upcoming challenges and future prospects that lie ahead. For the purpose of performing this systematic review, the suggested PRISMA guidelines were adhered to [13].

## 2. Brief Overview of AI Techniques Commonly Used in UIA Rupture Risk Assessment

Machine Learning (ML) techniques that rely on pre-extracted features and Deep Learning (DL) techniques that employ neural networks are the commonly used AI technologies in the risk assessment of IAs [14]. ML empowers AI with the capability to acquire knowledge and train models to extract and store features and their associated parameters. Especially suitable for various selective features that are nonlinearly related [15]. There exist three distinct types of ML: Supervised, Unsupervised, and Semi- or Weakly Supervised learning. Supervised learning involves training models using precisely labeled data or annotations. On the other hand, unsupervised learning entails training models without precise labels, wherein the algorithm clusters data to reveal underlying patterns. Lastly, semi- or weakly supervised learning employs both labeled and unlabeled data for training, thereby reducing the burden of annotation [16]. DL allows us to retrieve data features directly from neural networks by creating deep-level networks and performing adaptive feature extraction. It is a subset of ML that learns its key features from data without using explicit examples [17].

Traditional ML algorithms typically depend on predefined structures that explicitly describe the inherent patterns present in the data, as well as the regions of interest that are informed by professional knowledge and incorporate explicit parameters. Some examples of classical machine learning methods include random forests [18], K-nearest neighbor [19], and Support Vector Machine (SVM) [20]. These algorithms have been effectively applied in AI studies [21]. DL, on the other hand, differs from traditional ML algorithms in that it utilizes an artificial neural network to automatically extract information from images and generate its own filters, referred to as feature maps. Additionally, DL models can memorize visual patterns with the highest frequency. These models typically include pooling and convolution layers, as well as fully connected and normalization layers. Pooling layers can reduce the number of parameters and prevent overfitting. During the training process, the model adjusts the weights of the input until the best performance is achieved when the final output is compared to the actual data. This process can be repeated multiple times to improve performance [22]. Convolutional Neural Networks (CNNs) [23] are a common type of deep learning architecture. The basic idea behind CNNs is that each convolution layer contains a large number of convolutional kernels, which function as feature extractors. The feature map that is created after the convolution cores pass through the entire image contains a significant amount of the image’s information. Layer by layer abstraction is performed between layers using pooling and nonlinear activation procedures. Each layer can take the features it has learnt in the preceding layer and extract additional, more abstract features from them [22].

## 3. AI in The Rupture Risk Assessment of IAs

As revealed by a 10-year retrospective cohort research based on 258 UIAs, enlarged UIAs ruptured at a rate that was twelve times higher than non-enlarged UIAs [24]. Just like claimed by a prospective cohort research based on 368 UIAs in 2016, the rupture rate from aneurysm growth to therapy before the rupture was approximately 6.3% per aneurysm per year [25]. The aforementioned studies demonstrate that follow-up observation is a trustworthy and safe method for low-risk UIA, but once aneurysm enlargement is discovered, it must be treated immediately. Aneurysm enlargement can also replace aneurysm rupture as an observation index for risk assessment [26]. Because risk factors play a highly interacting role in UIA rupture and enlargement, related studies typically do not address rupture and enlargement separately.

According to the available research, the main analysis inputs for the intelligent prediction model are morphological data and hemodynamic variables. Different algorithms are also used, including CNN (one of the feedforward neural networks), SVM, and RC. In many cases, the dataset is divided into two subsets: the training set and the validation set. The ratio of stable and unstable instances in these subsets is usually kept as constant as possible. The training set is used to process the input data and develop a predictive model, while the validation set is used to evaluate the performance of each ML model. This evaluation typically includes metrics such as accuracy, sensitivity, and specificity to determine the general effectiveness of the model. During the analysis process, the weight value of each variable can be determined. Figure 1 depicts the general application model of artificial intelligence in the rupture risk assessment of IAs.

In a previous study, Liu et al. [31] utilized a two-layer feedforward Artificial Neural Network (ANN) to analyze 594 aneurysms (54 unruptured and 540 ruptured) and estimate their likelihood of rupture. The authors trained the ANN on 17 aneurysm features, which included 13 morphological characteristics of anterior communicating artery aneurysms identified in CT angiography images, as well as two demographic factors and histories of hypertension and smoking. To enhance the effectiveness of the neural network, the authors used an adaptive synthetic sampling strategy to create additional synthetic data of unruptured aneurysms. The areas under the Receiver Operating Characteristics (ROC) curve for the training, validation, testing, and total datasets were 0.953, 0.937, 0.928, and 0.950, respectively. The overall prediction accuracy for the 594 samples was 94.8%. However, the study was limited by its small, single-center Chinese population and the uneven sample size of ruptured and unruptured aneurysms, which may have limited the generalizability of the findings. Additionally, due to the lack of long-term follow-up data, it was difficult to track how aneurysms changed over time. Finally, further validation using separate patient data is necessary to validate the model’s effectiveness.

A recent study using data from Korea’s National Health Screening Program proposed a method for predicting aneurysm development. The study [32] utilized 21 basic clinical parameters and several ML techniques to develop and test four distinct predictive models. The models predicted the development of IAs with 75–77% accuracy, with minor variations in sensitivity, specificity, and incidence prediction across different ML algorithms. This method shows promise for clinical screening of aneurysms using basic clinical data, without the need for complex contrast-enhanced angiographic studies or costly magnetic resonance imaging. However, further improvement in the models’ accuracy is necessary before they can be used as a reliable screening tool. In this study, only clinical factors were utilized to create the model, and morphological information and hemodynamic variables could have been included to increase its performance stability. Additionally, testing these models on different populations is crucial for validating their effectiveness.

The research involved 364 UIAs training and 93 test datasets, and Ahn et al. [33] developed a multi-view CNN based on 3D-DSA to accurately predict the risk of unruptured intracranial aneurysm (UIA) rupture (high vs. low). They compared the performance of the multi-view CNN-ResNet50 to a single-image-based CNN (single-view ResNet50) and several other CNN architectures (AlexNet and VGG16) with various layers (ResNet101 and ResNet152). The models were evaluated and compared based on their sensitivity, specificity, and overall accuracy in risk prediction. The study found that the multi-view CNN-ResNet50 had the highest overall accuracy of 81.72%, with a sensitivity of 81.82% and a specificity of 81.63%, while the single-view CNN-ResNet50 had a lower sensitivity and overall accuracy compared to the multi-view CNN-ResNet50. The specificity of AlexNet, VGG16, ResNet101, and ResNet152 was similar to that of the multi-view CNN-ResNet50.

Kim et al. [34] analyzed DSA images in three dimensions of 368 cases of anterior circulation aneurysms with the largest diameter (<7 mm) and developed a computer-aided small aneurysm rupture risk prediction platform using CNN. Images of the patient’s aneurysms were taken in six different directions, and each image’s region-of-interest was then extracted. During prospective testing on 272 patients, the resulting CNN’s sensitivity, specificity, overall accuracy, and receiver operating characteristics were contrasted to those of a human assessor. In terms of predicting aneurysm rupture, the platform demonstrated a sensitivity of 78.76% (95% CI: 72.30–84.30%), a specificity of 72.15% (95% CI: 60.93–81.65%), and an overall diagnostic accuracy of 76.84% (95% CI: 71.3–81.72%). The CNN’s AUROC was 0.755 (95% CI: 0.699–0.805%), which was superior to the outcomes from a human assessor (AUROC: 0.537; *p* < 0.001). The authors came to the conclusion that the CNN-based prediction system could predict the possibility of a small aneurysm rupturing and had a higher diagnostic accuracy than human assessors. More big-data-based research is required to increase diagnostic precision and hasten practical applicability.

Similarly, with a sample pool of 528 unstable aneurysms (defined as ruptured within 1 month or growth during follow-up) and 1539 stable aneurysms (defined as unruptured, non-growing aneurysms), Zhu et al. [35] compared 13 types of clinical features and 18 morphological characteristics of aneurysms using three various AI models, including RC, SVM, and feedforward neural network—utilized for learning purposes—and compared the training results with traditional logistic regression analysis and PHASES score to predict aneurysm rupture. It was discovered that the AI model’s AUC value was significantly higher than that of the PHASES score (0.831–0.851 vs. 0.615, *p* < 0.01) and the conventional logistic regression analysis model (AUC = 0.810, *p* = 0.038). The efficacy outperformed conventional scoring systems and statistical models. This study, which is more in accordance with the current clinical diagnosis and treatment decision-making paradigm, not only analyzed the differences between ruptured and unruptured aneurysms, but also, for the first time, examined aneurysm growth in the analysis of aneurysm stability.

Detmer et al. studied the epidemiological, morphological, and hemodynamic variables of 1631 cases of aneurysms using K-nearest neighbor, SVM, RC, and other AI technologies, and ultimately achieved 76.0–79.0% accuracy of aneurysm rupture prediction [36]. Hemodynamics is a crucial evaluation component for determining the stability of aneurysms [37], and this study includes hemodynamic factors to improve the systematization of the model that was produced. However, the shortcomings of this study are also obvious. Firstly, the study population was limited to patients who underwent angiographic imaging, which may have introduced selection bias as patients who died from ruptured aneurysms before being admitted to the hospital or those who only underwent MRA or CTA were not included. Secondly, the training and test data were obtained from different populations and the vessel lumen segmentation for CFD simulations was conducted by different researchers using different methods. This may have introduced variability in the data, affecting the accuracy of the models. Additionally, the models were based on the assumption that aneurysms at high risk of rupture share similarities with those that have already ruptured, which has yet to be validated. Therefore, further studies are needed to confirm the reliability of these models in predicting the likelihood of future aneurysm rupture.

Additionally, the viability of using ML to forecast aneurysm stability was investigated by Liu et al. [38]. The study collected morphological data from 719 aneurysms using PyRadiomics, identifying 12 features. Of these, 420 aneurysms with maximum 3D diameters between 4 mm and 8 mm were selected for analysis, and their medical records were examined for durability and other clinical characteristics. ML models were developed to identify morphological determinants and compare the effectiveness of models with and without clinical features in predicting aneurysm stability. Additionally, the influence of clinical characteristics on the form of unstable aneurysms was investigated. The study concluded that the model’s area under the curve was 0.853 (95% CI, 0.767–0.940). Low values of Sphericity (*p* = 0.035), Flatness (*p* = 0.010), Compactness1 (*p* = 0.035), Compactness2 (*p* = 0.036), and Compactness (*p* = 0.035), but high Spherical Disproportion (*p* = 0.034), were observed in unstable aneurysms of hypertensive individuals. The research team innovatively utilized PyRadiomics to extract morphological features for aneurysm layering and proposed that flatness is the most critical morphological determinant for predicting aneurysm stability. The ML model developed in the study can be applied for this purpose.

A Back-Propagation (BP) neural network can be used to evaluate the risk of IA rupture/growth, according to Yang et al. [39]; thirthy-six people from a prospective registry research (ChiCTR190002447) who shared 45 features with 37 IAs were included in the study. All patients were monitored for 36 months following IA diagnosis or until aneurysm ruptured or grew. Clinical, morphological, and hemodynamic aspects were studied using multidimensional data. The analysis of hemodynamics was performed using patient-specific models. The study created seven BP neural network models using various traits, with a training set to validation set ratio of 8:1. Yang’s team found that, among the models that determined three-year IA stability based on a single variable of IA parameters, only morphological characteristics performed well (AUC = 0.703, *p* = 0.035). The likelihood of three-year IA rupture/growth was predicted by the clinical–morphological (AUC = 0.731, *p* = 0.016), clinical–hemodynamic (AUC = 0.702, *p* = 0.036), and morphological–hemodynamic (AUC = 0.785, *p* = 0.003) models that combined two dimensions of IA features. The models that incorporated all three dimensions displayed the highest predictive power (AUC = 0.811, *p* = 0.001). However, the study had several limitations. Due to the small sample size, no testing set was available to evaluate the network’s performance after training. The sample sizes of the stable and unstable IA groups were balanced, resulting in only 7% of the overall sample consisting of patients in the unstable IA group. Additionally, the morphological traits were manually assessed by skilled medical professionals, which may have introduced subjectivity bias.

Recently, 104 patients with small IAs from two centers were retrospectively analyzed by Xiong et al. [40] to develop a ML model allowing simple external validation to assess the rupture risk of small IAs. To divide the patients at Center 1 into a training set (70%) and an internal validation set (30%), random stratification was employed. The patients at Center 2 were used as the external validation set. The researchers constructed five ML models using various algorithms, namely the RFC, SVM with linear kernel, categorical boosting (CatBoost), light gradient boosting machine (LightGBM), and extreme gradient boosting. The least absolute shrinkage and selection operator (LASSO) method (XGBoost) was used to choose predictive features. The best ML model was explained using the Shapley Additive Explanations (SHAP) study. In terms of performance, the SVM model outperformed the PHASES score significantly (all *p* < 0.001), with an AUC of 0.817 in the internal validation cohorts (95% CI, 0.769–0.866) and 0.893 in the external validation cohorts (95% CI, 0.808–0.979). Maximum size, position, and irregular form were the three most crucial characteristics for rupture prediction, according to SHAP analysis. In this study, rupture risk of small-sized IAs could be predicted by the SVM model with satisfactory discriminative capacity.

In general, virtually all existing research claims that the rupture risk prediction or instability risk prediction model has higher performance than the PHASES score and other regularly used risk assessment methods in the past, so that the design of treatment regimens can be more evidence-based.

## 4. Challenges

AI studies aimed at assessing the risk of rupture in IA have several common limitations. One such limitation is the use of relatively small samples for both training and validation in current studies investigating the use of AI for predicting aneurysm rupture risk. To enhance AI prediction performance, further prediction models may think about using more samples and merging more pertinent parameters. Then, multi-center randomized controlled trials are required for additional validation of superior predictive ability. The majority of the currently used intelligent prediction models are retrospective, single-center studies with variable case quality, which raises concerns about selection bias and a lack of consistency in imaging procedures. The algorithms suggested for the same task have inconsistent performance. It can be attributed to several factors, including variations in the training data, the machine learning approach utilized, the use of image-based features, and the image processing techniques employed (e.g., imaging techniques, lesion sizes, and locations). Therefore, to achieve repeatable outcomes and consistent performance, prospective research with standardized protocols is necessary [41]. Kim et al. suggested using the following criteria to determine whether AI is useful in clinical settings: (1) must acquire external validation; (2) must employ diagnostic cohort studies; (3) must use high-quality data from numerous institutions; and (4) must undertake prospective research [41].

Furthermore, many studies in this field also suffer from a lack of a diagnostic gold standard reference, which can have an indirect impact on the algorithm’s performance and lead to overly optimistic results that are not necessarily based on reliable ground truth. On the other hand, since DSA is expensive and invasive, it is often only used for specific clinical indications, and retrospective inclusion of only DSA-verified studies could introduce bias in the results [1,42]. This challenge could be partly addressed through a prospective implementation of the gold standard test, either by means of standardized double-reading radiological diagnosis by specialists in the field or through random administration to the targeted population.

A possible hazard and ethical concern of AI-based systems has also been brought up in the literature: automation bias [43,44,45]. Automation bias refers to the reader’s reliance on the machine and their omission of opposing information created without automation, which lowers their self-confidence and eliminates human contribution throughout the interpretation process and may eventually reduce their sensitivity. According to a systematic study, measures to reduce automation bias should focus on lowering the cognitive burden. The degree of cognitive load experienced during decision-making processes is related to automation bias [43].

## 5. Future Perspectives

After being integrated with the medical industry, AI technology, which is a cutting-edge new technology, has a bright future in the rupture risk prediction of IAs. Future prospects include the addition of new projects and the coordination of the procedures that are already in place [46]. Along with the aneurysms, various vascular abnormalities (such as stenosis, arteriovenous malformation, and Moyamoya disease) may be assessed using AI-based methods. Advanced network architectures and training methods, such as convolutional residual networks, active learning, one-shot learning, and generative adversarial networks, may help overcome some of the current limitations in AI-based risk assessments of intracranial aneurysm rupture. These approaches are currently under active investigation and require further research to fully understand their potential and limitations [47,48].

## 6. Conclusions

Artificial intelligence could be used to evaluate the rupture risk of cerebral aneurysms. In the future, we hope to apply it to treat patients with aneurysms, scientifically and effectively. Applications that are effective mostly concentrate on the detection and evaluation of hemodynamics and morphology. The current research on AI for IAs is limited due to its small sample size, retrospective design, focus on comparison rather than prediction, and inconsistent data quality. Therapeutic implications are also limited due to variable performance, high false-positive rates, system complexity, and cost. To overcome these limitations and improve the clinical applicability of AI in the investigation of IA, there is a need for large-scale prospective multi-center studies that utilize high-quality input data and aim to improve prediction accuracy.

## Figures and Tables

**Figure 1 brainsci-13-01056-f001:**
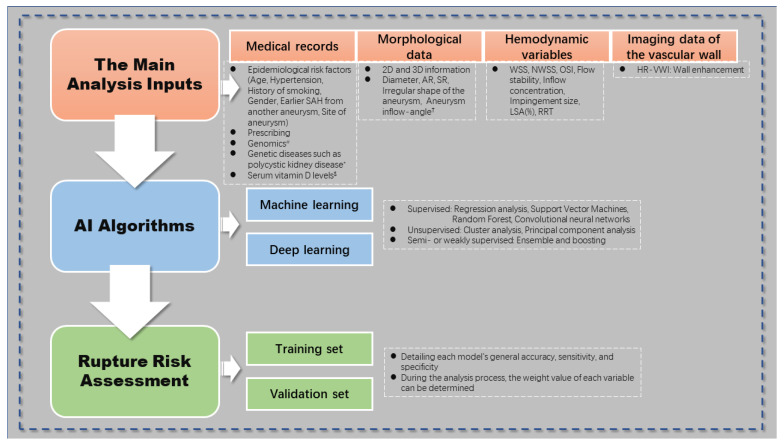
Guidelines for using artificial intelligence in intracranial aneurysm rupture risk assessment. **#** In terms of genomics, studies have mentioned that three single nucleotide polymorphisms may be associated with aneurysms [27]. ***** Aneurysm risk is proportionately higher in those with genetic conditions like polycystic kidney disease [28], although it is yet unclear whether the risk of rupture is also higher. $ According to a study, patients with ruptured aneurysms had considerably decreased serum vitamin D levels [29]. † The angle between the parent artery’s central axis and the aneurysm’s main axis is known as the aneurysm inflow-angle. It is a significant risk factor for the rupture of lateral wall aneurysms, according to studies [30]. SAH, subarachnoid hemorrhage; AR, aspect ratio; SR, size ratio; WSS, wall shear stress; NWSS, normalized WSS; OSI, oscillatory shear index; LSA, low WSS area; RRT, relative residence time; HR-VWI, high resolution magnetic resonance vessel wall imaging.

## Data Availability

The data sets analyzed during the current study are available from the corresponding author on reasonable request.

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
