# Peer review of "A Review of Artificial Intelligence in the Rupture Risk Assessment of Intracranial Aneurysms: Applications and Challenges"

_brainsci, 2023, doi:10.3390/brainsci13071056_

Round 1
Reviewer 1 Report
This is an important review of the current role of AI in assessing the rupture risk of intracranial aneurysms. It included a succinct introduction on the methods and variations of machine learning. The limitations of each study and the overall challenge is very well covered. It as a very well presented review.
The review analyses more or less ten studies involving AI assessing un-ruptured aneurysms. The authors mentioned compliance with PRISMA in their systemic reviews. On what basis were these particular studies selected and whether there were other studies rejected, was not specified. If possible a brief accompanying table summarising these included studies, for example, with the number of cases, methods, findings and a comment may facilitate the data presentation.
The review is mainly on the rupture risk assessment. Perhaps, as the rationale of the study is after all regarding the decision-making and benefit:risk evaluation, a brief mention the other aspects of AI in assessing the individual treatment success and morbidity risk would be even more challenging.
Author Response
Response to Reviewer 1 Comments
Point 1: The review analyses more or less ten studies involving AI assessing un-ruptured aneurysms. The authors mentioned compliance with PRISMA in their systemic reviews. On what basis were these particular studies selected and whether there were other studies rejected, was not specified. If possible a brief accompanying table summarising these included studies, for example, with the number of cases, methods, findings and a comment may facilitate the data presentation.
Response 1:
Thank you for your valuable feedback on our manuscript. We appreciate your thoughtful comments and suggestions for improvement.
Regarding the selection of studies in our systematic reviews, we apologize for not providing specific details about the basis for inclusion and any rejected studies. We understand the importance of transparency in study selection and will address this in the revised manuscript. We will provide a clear and comprehensive explanation of the criteria used for study inclusion and highlight any excluded studies.
Additionally, we acknowledge your suggestion to include a brief accompanying table summarizing the included studies. We agree that this would enhance the presentation of the data and allow readers to have a quick overview of the number of cases, methods, findings, and any relevant comments. In line with your recommendation, we will include a concise table that summarizes the key characteristics of the included studies.
Once again, we appreciate your constructive feedback, and we will make the necessary revisions to address all your concerns. If you have any further suggestions or comments, please do not hesitate to let us know.
Point 2: The review is mainly on the rupture risk assessment. Perhaps, as the rationale of the study is after all regarding the decision-making and benefit:risk evaluation, a brief mention the other aspects of AI in assessing the individual treatment success and morbidity risk would be even more challenging.
Response 2:
Thank you for your insightful comments on our manuscript. We appreciate your suggestion to expand the scope of our review to encompass other aspects of AI in assessing individual treatment success and morbidity risk, in addition to rupture risk assessment.
We agree with your point that decision-making and benefit:risk evaluation are integral components of our study's rationale. Therefore, it would indeed be beneficial to briefly mention how AI can potentially contribute to assessing treatment success and morbidity risk in individual cases. By incorporating this aspect, we can provide a more comprehensive overview of the potential applications and challenges of AI in neurosurgical decision-making.
In the revised manuscript, we will include a subsection or paragraph dedicated to discussing how AI can be utilized beyond rupture risk assessment, specifically focusing on its potential role in evaluating individual treatment outcomes and predicting morbidity risks. We will provide relevant references and highlight the challenges associated with leveraging AI in these areas.
Once again, we appreciate your valuable input, and we will incorporate your suggestion into the revised manuscript. If you have any further recommendations or comments, please feel free to let us know.

Reviewer 2 Report
I was so glad to review this paper. It is crucial to predict the risk of rupture in intracranial aneurysm. however, the conventional methods or previous observation studies did not answer this issue. So all neurosurgeons wished for a believable answer to this issue.
The authors searched ten studies that analyzed a rupture risk of aneurysm using AI. They evaluated each study and pointed out the limitation of each study. And They suggested the way of the study.
I wished this study to be read by All neurosurgeons.
Author Response
Response to Reviewer 2 Comments
Point 1: I was so glad to review this paper. It is crucial to predict the risk of rupture in intracranial aneurysm. however, the conventional methods or previous observation studies did not answer this issue. So all neurosurgeons wished for a believable answer to this issue.
The authors searched ten studies that analyzed a rupture risk of aneurysm using AI. They evaluated each study and pointed out the limitation of each study. And They suggested the way of the study.
I wished this study to be read by All neurosurgeons.
Response 1:
Thank you for your positive review of our paper. We are delighted to hear that you found the topic of predicting the risk of rupture in intracranial aneurysms to be crucial and that our study addresses an important gap in the existing literature.
We understand the need for a reliable approach to answer this significant issue, especially considering the limitations of conventional methods and previous observation studies. Our aim was to provide neurosurgeons with a comprehensive evaluation of ten studies that utilized AI to analyze the rupture risk of aneurysms. By critically assessing each study and highlighting their limitations, we aimed to contribute to the development of future research in this field.
Your endorsement and desire for our study to reach all neurosurgeons is greatly appreciated. We hope that our findings and recommendations will indeed benefit the neurosurgical community by providing valuable insights and directions for further investigation.
Once again, we sincerely thank you for your positive feedback and support. Should you have any additional suggestions or comments, please do not hesitate to let us know.

Reviewer 3 Report
Dear Authors,
I appreciate the opportunity to review Your work. The article is professionally written, consistent. The topic is indeed important and AI is becoming an everyday tool for Medical workers. The article covers the most important papers considering aneurysms and AI. Although the manuscript is well writen, I do not see much of clinical or educational vaule as similar papers on this topic have been published in the recent years with a better evaluation of results and for instance comparison of algorithm results.
Regards,
Reviewer
Author Response
Response to Reviewer 3 Comments
Point 1: I appreciate the opportunity to review Your work. The article is professionally written, consistent. The topic is indeed important and AI is becoming an everyday tool for Medical workers. The article covers the most important papers considering aneurysms and AI. Although the manuscript is well writen, I do not see much of clinical or educational vaule as similar papers on this topic have been published in the recent years with a better evaluation of results and for instance comparison of algorithm results.
Response 1:
Thank you for taking the time to review our work and for your positive feedback on the quality of our writing and the importance of the topic. We appreciate your acknowledgment of the relevance of AI as an everyday tool for medical professionals, particularly in the context of aneurysms.
We also understand your concern regarding the clinical and educational value of our manuscript compared to similar papers published in recent years. We acknowledge that previous studies may have provided better evaluations of results and included comparisons of algorithm outcomes. In light of this feedback, in our revised manuscript, we will aim to enhance the clinical and educational value by providing a more comprehensive evaluation of the results and incorporating relevant comparisons with other algorithms where applicable.
We appreciate your constructive criticism, and we are committed to improving our manuscript based on your suggestions. If you have any further specific recommendations or comments, we would be grateful to hear them.
Thank you once again for your time and input.

Reviewer 4 Report
The review is well written, has a very important subject.
The authors wrote a brief review of artificial intelligence's role in the assessment of rupture risks of intracranial aneurymsms. This is a goog contribution to the field that gives an overview of AI techniques commonly used in unruptured intracranial aneurysms (UIAs) and its role in the rupture risk assessment, challenges and future perspectives. The Introduction part is adequately described with crucial information. The Results, Discussion and Conclusion parts are consistent and adequately described. The references are appropriate.The authors have included the guidelines for using artificial intelligence in intracranial aneurysm rupture risk assessment. I only recommend one more figure or scheme for readers.
My censorious advice is to make one more figure to make this review even more attractive to readers.
Author Response
Response to Reviewer 4 Comments
Point 1:
The review is well written, has a very important subject.
The authors wrote a brief review of artificial intelligence's role in the assessment of rupture risks of intracranial aneurymsms. This is a goog contribution to the field that gives an overview of AI techniques commonly used in unruptured intracranial aneurysms (UIAs) and its role in the rupture risk assessment, challenges and future perspectives. The Introduction part is adequately described with crucial information. The Results, Discussion and Conclusion parts are consistent and adequately described. The references are appropriate.
The authors have included the guidelines for using artificial intelligence in intracranial aneurysm rupture risk assessment. I only recommend one more figure or scheme for readers.
My censorious advice is to make one more figure to make this review even more attractive to readers.
Response 1:
Thank you for your positive feedback on our manuscript and for recognizing the importance of the subject matter. We are glad that you found our review of artificial intelligence's role in assessing rupture risks of intracranial aneurysms to be a valuable contribution to the field.
We appreciate your suggestion regarding the inclusion of an additional figure or scheme to further enhance the attractiveness of the review for readers. However, after careful consideration, we have decided not to add any more figures to the manuscript at this time.
While we understand the potential benefits of visual aids in enhancing reader engagement, we believe that the current text adequately conveys the key concepts and insights. We have taken great care to ensure that the information is presented clearly and comprehensively without the need for additional visual representations.
We genuinely value your input and take all suggestions into careful consideration. We are grateful for your understanding regarding our decision not to include another figure.
Once again, we sincerely appreciate your time and effort in reviewing our manuscript.
